# Mapping Paddy Rice with Satellite Remote Sensing: A Review

**Rongkun Zhao [1] , Yuechen Li [1,2,\*] and Mingguo Ma [1,2]**

1 Chongqing Jinfo Mountain Karst Ecosystem National Observation and Research Station, School of Geographical Sciences, Southwest University, Chongqing 400715, China; zrk1998@email.swu.edu.cn (R.Z.); mmg@swu.edu.cn (M.M.)

2 Chongqing Engineering Research Center for Remote Sensing Big Data Application, School of Geographical Sciences, Southwest University, Chongqing 400715, China

\* Correspondence: liyuechen@swu.edu.cn; Tel.: +86-23-682-53912

**Abstract:** Paddy rice is a staple food of three billion people in the world. Timely and accurate estimation of the paddy rice planting area and paddy rice yield can provide valuable information for the government, planners and decision makers to formulate policies. This article reviews the existing paddy rice mapping methods presented in the literature since 2010, classifies these methods, and analyzes and summarizes the basic principles, advantages and disadvantages of these methods. According to the data sources used, the methods are divided into three categories: (I) Optical mapping methods based on remote sensing; (II) Mapping methods based on microwave remote sensing; and (III) Mapping methods based on the integration of optical and microwave remote sensing. We found that the optical remote sensing data sources are mainly MODIS, Landsat, and Sentinel-2, and the emergence of Sentinel-1 data has promoted research on radar mapping methods for paddy rice. Multisource data integration further enhances the accuracy of paddy rice mapping. The best methods are phenology algorithms, paddy rice mapping combined with machine learning, and multisource data integration. Innovative methods include the time series similarity method, threshold method combined with mathematical models, and object-oriented image classification. With the development of computer technology and the establishment of cloud computing platforms, opportunities are provided for obtaining large-scale high-resolution rice maps. Multisource data integration, paddy rice mapping under different planting systems and the connection with global changes are the focus of future development priorities.

**Keywords:** optical remote sensing; microwave remote sensing; phenology-based method

## 1. Introduction

Paddy rice, as a major staple food, feeds almost half the world's population [1]. As the population grows, the demand for food grows. In terms of water use, about one-quarter to one-third of the world's freshwater resources are used for paddy rice irrigation [2]. Paddy rice fields are a major source of methane ($CH_4$) emissions [3]. Globally, methane ($CH_4$) emissions from paddy rice account for more than 10% of the total amount of $CH_4$ in the atmosphere [4]. Methane is the second most abundant greenhouse gas after carbon dioxide [5]. Paddy rice fields serve as habitats for birds, ducks, and other species, which are the origin of highly pathogenic avian influenza [6]. Therefore, the development of paddy rice distribution maps is of great significance for understanding and assessing the environmental conditions of food security, climate change, disease transmission and water use at regional, national and global levels [7].

An in-depth understanding of paddy rice cultivation and physiology is the premise of paddy rice mapping. The general physical characteristics of different crops are different, and the characteristics of paddy rice at different growth stages are also different. The paddy rice growth period can be divided into four stages [7]: (1) from sowing to transplanting in the nursery stage (~1 month), (2) from the transplanting to the heading

stage (1.5 to 3 months), (3) from the heading to the reproductive stage with flowering (~1 month, including start, heading and flowering, stem elongation and panicle development), and (4) from flowering to mature stages at full maturity (~1 month, including milk stage, dough, and ripe grains). The morphology of paddy rice at the main growth stages is shown in Figure 1. Paddy rice is the only crop that needs extensive water during the growing phase and is the only staple that needs transplanting. Therefore, paddy rice can be identified by studying the sensitive spectral bands or indices during the period of water, soil, and vegetation mixing. Temporal variation in water–soil–vegetation composition is a key factor in paddy rice identification.

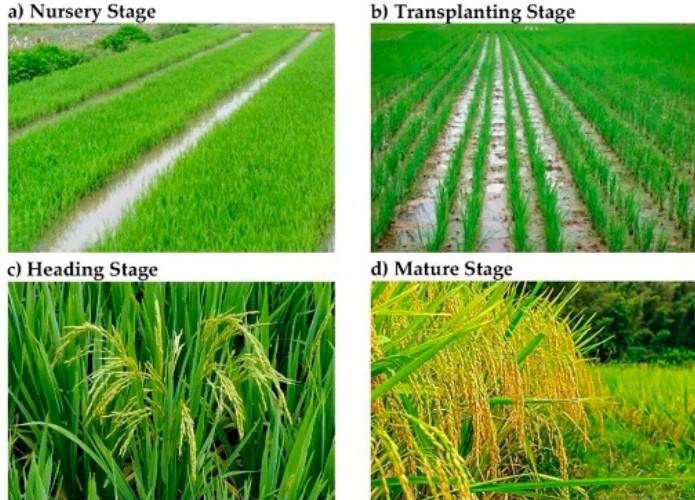

**Figure 1.** The example of paddy rice growth stages.

In previous research, some scholars summarized and analyzed the content related to rice mapping. Dong et al. [7] discussed the evolution of rice mapping methods from the 1980s to 2015 and summarized the methods used to characterize each stage and future development trends. Claudia et al. [1] mainly discussed the basic work of rice mapping. Based on a large number of studies, they summarized the characteristics of rice mapping (such as sensors, vegetation index, biomass) and summarized the application fields of different satellite sensors. Mostafa et al. [8] discussed the applicability of remote sensing images to rice area mapping and yield prediction. Methods and limitations of mapping and yield prediction using different remote sensing sensors are briefly described. Niel et al. [9] mainly discussed the current status of the application of remote sensing technology in rice planting areas in Australia, including crop identification, area measurement, and yield prediction. The research of these scholars is of great significance to the understanding of rice mapping methods. However, there are also some deficiencies: (I) The systematic induction of rice mapping methods is not comprehensive enough. (II) Detailed introductions of the method principles are limited. (III) There is a lack of comparison and evaluation of different methods. Based on the above analysis, this paper conducts a systematic review of rice mapping methods under recent remote sensing technology to correctly select rice mapping methods suitable for specific research purposes.

Paddy rice mapping algorithms are diverse. These methods include supervised classification and unsupervised classification methods, phenology algorithms, and object-oriented image classification. Data sources include optical remote sensing and microwave remote sensing. To facilitate the subsequent development of new methods, this study reviews paddy rice mapping methods in the literature since 2010. The content mainly include three aspects: (I) Methods based on optical remote sensing data and their advantages and disadvantages; (II) Methods based on microwave remote sensing data and their advantages and disadvantages; (III) Methods based on the integration of optical remote sensing data and microwave remote sensing data and their advantages and disadvantages. Finally,

we summarize the development trend of paddy rice mapping methods, as well as the challenges and future direction of development.

## 2. Key Feature Statistics in Remote Sensing

We used Web of Science to collect rice-related papers published in major international remote sensing journals and some agricultural journals between 2010 and 2020. The search keywords were rice, remote sensing, and mapping. Afterwards, further screening was carried out by reading the abstract, and highly relevant papers were selected for review and analysis.

The main data sources are shown in Table 1. The most commonly used optical remote sensing satellites are Landsat, MODIS and HJ-1A/B, with spatial resolutions of 30 m and 500 m, respectively. In recent research on radar data, Sentinel-1 data are often used, with a spatial resolution of 10 m and temporal resolution less than 10 days. In addition, we made statistics on the relevant characteristics of the data sources in the integration method. It is worth mentioning that the integration method is mostly based on Landsat data set in optical images, extracting all high-quality images in a period of time for research, and the time resolution is 8 days~16 days (Table 2). These satellite sensors have the potential to obtain multitemporal and multispectral reflectance data on farmland. These data can be used to derive the time series of vegetation indices (VIs), which are calculated as a function of the red, green, blue, and infrared spectral bands (see the major VIs in Table 3).

**Table 1.** Main satellite data sources used for paddy rice mapping.

| Satellite | Sensor | Spatial Resolution | Temporal Resolution | Free or Charge | Literature Number |
|---|---|---|---|---|---|
| Landsat | MSS+TM (Landsat-5) ETM+ (Landsat-7) OLI (Landsat-8) | 30 m | 16 days | Free | 16 |
| Terra/Aqua | MODIS | 250–1000 m | 1–2 days | Free | 22 |
| HJ-1A/B | CCD1/2 | 30 m | 2–4 days | Free | 3 |
| SPOT | HRV (SPOT1~3) VGT (SPOT-4) HRG/HRS/VGT (SPOT-5) | 1 km | 1 day | Charge | 2 |
| Sentinel-2 | MSI | 10–20 m | 5 day | Free | 7 |
| Sentinel-1 | SAR | 5–40 m | 12 days | Free | 14 |
| COSMO-SkyMed | SAR | 3–15 m | 16 days | Charge | 1 |
| TerraSAR-X | SAR | 3–10 m | 11 days | Charge | 1 |
| ENVISAT | ASAR | 20–500 m | 35 days | Free | 2 |
| RADARSAT-1 | SAR | 10–100 m | 24 days | Charge | 1 |
| RADARSAT-2 | SAR | 3–100 m | 24 days | Charge | 2 |
| ALOS-2 | PALSAR-2 | 25 m | 14 days | Charge | 3 |

**Note:** MSS—Multispectral Scanner; TM—Thematic Mapper; ETM+—Enhanced Thematic Mapper Plus; OLI—Operational Land Imager; CCD—Charge Coupled Device; HRV—High Resolution Visible; VGT—VEGETATION; HRG—High Resolution Geometric Imaging Instrument; HRS—High Resolution Stereoscopic Imaging Instrument; MSI—Multi-Spectral Instrument; SAR—Synthetic Aperture Radar; ASAR—Advanced Synthetic Aperture Radar; Literature Number—The number of literature using this data in the literatures included in this study.

**Table 2.** Introduction to Integration Method Data Sources.

| Integrated Data Sources | Integrated Spatial Resolution | Integrated Time Resolution | Ref. |
|---|---|---|---|
| Landsat ETM+\OLI | 30 m | 8 days | [10] |
| Landsat 8 OLI MODIS | 30 m | 16 days | [11] |
| Landsat TM\ETM+\OLI | 30 m | <16 days | [12] |
| Landsat TM\ETM+ | 30 m | ≤16 days | [13,14] |
| Landsat ETM+\OLI | 30 m | 16 days | [15] |
| Sentinel-2 MODIS | 10 m | 16 days | [16] |

**Table 3.** Indices used in paddy rice mapping methods.

| Index | Abbreviation | Formula | Literature Number |
|---|---|---|---|
| Normalized Difference Vegetation Index | NDVI | $\frac{\rho_{nir}-\rho_{red}}{\rho_{nir}+\rho_{red}}$ | 27 |
| Enhanced Vegetation Index | EVI | $2.5 \times \frac{\rho_{nir}-\rho_{red}}{\rho_{nir}+6\times\rho_{red}-7.5\times\rho_{blue}+1}$ | 14 |
| Two-band Enhanced Vegetation Index | EVI2 | $2.5 \times \frac{\rho_{nir}-\rho_{red}}{\rho_{nir}+2.4\times\rho_{red}+1}$ | 1 |
| Land Surface Water Index | LSWI | $\frac{\rho_{nir}-\rho_{swir}}{\rho_{nir}+\rho_{swir}}$ | 16 |
| Normalized Difference Snow Index | NDSI | $\frac{\rho_{green}-\rho_{swir}}{\rho_{green}+\rho_{swir}}$ | 3 |
| Normalized Difference Water Index | NDWI | $\frac{\rho_{green}-\rho_{nir}}{\rho_{green}+\rho_{nir}}$ | 1 |
| Normalized Difference Flood Index | NDFI | $\frac{\rho_{swir}-\rho_{red}}{\rho_{swir}+\rho_{red}}$ | 1 |

**Note:** Surface reflectance values from the blue ($\rho_{blue}$), green ($\rho_{green}$), red ($\rho_{red}$), Near Infrared (NIR) ($\rho_{nir}$) and Shortwave Infrared(SWIR) ($\rho_{swir}$) bands.

## 3. Taxonomy

In the past 10 years, scholars have carried out many studies on the paddy rice mapping method and further improved the method's precision based on its predecessors. The methods can be divided into the following three categories, based on the difference in data sources: optical remote sensing mapping method, microwave remote sensing mapping method, and integrated method (Table 4). According to the method principle, the optical remote sensing mapping method is further divided into four categories. Generally, the microwave remote sensing mapping method extracts the variation in the backscattering coefficient of paddy rice during the growing period, which can be divided into two categories according to the principle of the method. The methods of optical and microwave integration are divided into two categories according to the principle of the method: complementary method and comparative method. Among the above methods, time series analysis methods such as phenology-based methods are relatively common, and object-oriented, deep learning, and data integration methods have been relatively innovative in the past 10 years.

**Table 4.** Categories of existing paddy rice mapping methods.

| Main Classification | Specific Methods | Refs. |
|---|---|---|
| Optical Remote Sensing-Based Mapping Methods | Machine learning | [13,17–23] |
| | Time series similarity method | [24,25] |
| | Vegetation index feature-based method | [10–12,14,26–41] |
| | Object-based image analysis | [42–44] |
| Microwave Remote Sensing-Based Mapping Methods | Empirical model | [45–48] |
| | Machine learning | [49–58] |
| Integration of Optical and Microwave Remote Sensing-Based Mapping Methods | Complementary method | [16,59–65] |
| | Comparison class method | [66–69] |

## 4. Evolution of Paddy Rice Mapping Methods

Remote sensing platforms can repeatedly observe the Earth's surface and collect a variety of data, so several remotely based methods have been developed to map paddy rice areas around the world. There are three types of methods based on different data sources. These methods are described in the following sections.

### 4.1. Optical Remote Sensing-Based Mapping Methods

Optical remote sensing sensors have been used extensively for mapping paddy rice areas around the world. The earliest method of paddy rice monitoring was to extract paddy rice by using remote sensing images and classification methods. Later, with the emergence of the vegetation index, phenological algorithm, cloud computing, and machine learning, the precision of paddy rice mapping based on optical remote sensing was constantly improved.

#### 4.1.1. Machine Learning

Machine learning methods are commonly used methods of rice mapping, including traditional machine learning and deep learning. Traditional machine learning includes supervised and unsupervised classification, such as ISODATA, decision tree (DT), random forest (RF), support vector machine (SVM). The principle of this type of method is to first collect images and sample training data and determine the decision rules to extract rice based on characteristic parameters.

Supervised classification is based on the samples provided by the known training area to obtain feature parameters to establish decision rules. Unsupervised classification obtains feature parameters through computer agglomeration statistical analysis of images to establish decision rules. The latter is an image classification method without a priori classification criteria. The input into the classifier is mainly preprocessed spectral images [13,21]. In recent years, normalized difference vegetation index (NDVI) temporal curves have also been used as the characteristic parameter for classification [18]. Manjunath et al. [17] used multitemporal SPOT VGT NDVI data for analysis. The ISODATA clustering method is used to distinguish between paddy rice areas and non-paddy rice areas. Then, the auxiliary data set is used to further subdivide the areas. Similarly, Okamoto et al. [13] used this method to extract paddy rice fields in Heilongjiang. The difference is that this study used Landsat TM/ETM+ as the data source. Gumma et al. used MODIS data products combined with the k-means clustering algorithm to map the paddy rice area. In 2011 and 2015, they used the same method to map paddy rice in different regions [18,19]. The results were relatively good, with a correlation of more than 90% with local statistics. Paddy rice mapping is also carried out using supervised classification methods, such as SVM [20] and RF [21]. The advantage of this method is its strong operability. The basic principles are easy to understand. The difficulty of the supervised method may be the collection of training samples. However, Google Earth's high-resolution images and the global geo-referenced field photo library (http://eomf.ou.edu/photos/) provide convenience. The disadvantage of this method is that the validity of the image will affect the accuracy of the results.

For example, cloudy and foggy areas, broken terrain areas, and mixed pixel problems will affect the results. In addition, the threshold settings in supervised classification and unsupervised classification methods will change according to the study area.

Deep learning performs well in image recognition and signal processing. In optical remote sensing, the CNN method is mainly used. Convolutional neural network (CNN) is well applied in the field of image analysis. In terms of scene classification, the CNN algorithm has higher classification accuracy than traditional algorithms. CNN is composed of several layers with different functions: input layer, convolution layer, pool layer, fully connected layer and output layer. The input layer is used to import training data, and the convolutional layer is used to extract features. The steps of applying this method to rice mapping are as follows: (1) Use the University of California Merced land-use data set, land-use/land-cover (LULC) Map, Google Earth high-resolution images, and field survey data to pretrain the model. (2) Input the original image into the model and output the result. In this step, in addition to the spectral data, NDVI, Land Surface Temperature (LST), and related phenological information can also be imported into the model [22]. Common training outputs classification results. Zhao et al. [23] combined CNN classification results with the results of NDVI under the DT to achieve further classification and output the final classification results.

The accuracy of this method is generally high. The overall accuracy is greater than 93%. The advantage is improved classification of complex surfaces and the broken landscapes. The disadvantage is that complex models require a lot of data for training. If the tagged data are not enough to support the entire training process, the deep learning model will have poor results. Therefore, the correct amount of training data guarantees the reliability and rationality of the training model.

### 4.1.2. Time Series Similarity Method

A new method that appeared in recent years is the time series similarity method of dynamic time warping (DTW) distance [24,25]. Time series similarity measures are used to describe the characteristics of data changes over time. DTW distance was initially applied to text data matching, speech processing, and visual pattern recognition. The research shows that algorithms based on the nonlinear bending technique can obtain high recognition and matching accuracy. The steps in this method are as follows: first, establish the standard NDVI sequence curve of the paddy rice growth cycle through field sample data, and then determine the threshold based on the DTW distance between the NDVI time series of standard paddy rice growth and the NDVI time series of the pixels to extract the paddy rice field. The principle of the time sequence similarity method based on the DTW distance is as follows:

Suppose two time series, i.e., $S_1(t) = \{s_1^1, s_2^1, \cdots, s_m^1\}$, $S_2(t) = \{s_1^2, s_2^2, \cdots, s_n^2\}$, with respective lengths of $m$ and $n$. Construct an $m \times n$ matrix $A_{m \times n}$ and define the distance between each element as $a_{ij} = d\left(s_i^1, s_j^2\right) = \sqrt{\left(s_i^1, s_j^2\right)^2}$. In the matrix $A_{m \times n}$, a winding path is set by a group of adjacent matrix elements, and notes for W = {w_1,w_2,···,w_k} and the $k_{th}$ element in W are defined as $w_k = \left(a_{ij}\right)_k$; this path meets the following conditions:

Monotonicity constraint: $w_k = a_{ij}, w_{k+1} = a_{i'j'}, i' \geq i, \ j' \geq j$,

Continuity constraint: $w_k = a_{ij}, w_{k+1} = a_{i'j'}, \ i' \leq i+1, \ j' \leq j+1$,,

Endpoint constraint: $w_1 = a_{11}, w_k = a_{mn}$.

This element satisfies the condition $0 \leq i - i', 0 \leq j - j' \leq 1$, and thus, $DTW(S_1, S_2) = min\frac{1}{K}\sum_{i=1}^{k} W_i$. The DTW algorithm can be summarized by applying ideal dynamic programming to find the best (i.e., least bending) cost path, as shown in Formula (1):

$$\begin{cases} D(1,1) = a_{11} \\ D(i,j) = a_{ij} + min\{D(i-1,j-1), D(i,j-1), D(i-1,j)\} \end{cases} \tag{1}$$

where $i$ = 2,3 . . . $m$, $j$ =2,3 . . . $n$, $D$ ($m$, $n$) is the minimum cumulative value of the winding paths.

The DTW distance can reflect the similarity and difference between the standard paddy rice growth NDVI time series and the NDVI time series of a pixel. In the DTW algorithm, when the DTW distance is short, the curve of the NDVI time series shows high similarity. We performed correlation analysis on the NDVI time series and ground truth data to determine the DTW distance threshold for identifying single- and multi-cropping paddy rice. Assuming that the DTW distance of the pixel is greater than the threshold shown, the pixel is unlikely to be paddy rice.

In 2014, Guan et al. [24] extracted rice areas from Southeast Asia and initially explored the applicability of this method in cloudy and rainy areas with good results. In 2018, the same team used this method to extract rice areas in Vietnam, and the results correlated well with statistical data ($R^2 = 0.809$). This result showed once again the potential of this method for rice mapping in monsoon regions and multiple cropping systems with diverse cultivation processes [25].

The accuracy of this method is good, and the overall accuracy is 83%. The advantage of this method is that it is suitable for cloudy and rainy areas, and the similarity analysis based on DTW distance can solve the overall curve deviation caused by the flexibility of paddy rice planting. This method has good application potential in different crops and different cropping systems. The disadvantages are the determination of the empirical model threshold and the determination of the NDVI standard curve. Affected by the spatial resolution of satellite data, the accuracy of the national scale is high and that of the provincial scale is low.

### 4.1.3. Vegetation Index Feature-Based Method

The third method is the vegetation index feature-based method. This method can be divided into two categories. One is the features are obtained through mathematical analysis. The threshold formula is established by mathematical analysis of the vegetation index time series curve. The other is the phenology algorithm. The principle is to extract paddy rice, which is grown on flooded soils, based on the unique physical characteristics. NDVI < Land Surface Water Index (LSWI) or Enhanced Vegetation Index (EVI) < LSWI during the flooding period of paddy rice, but the EVI value of other vegetation (non-flooded) is usually greater than the LSWI value.

Mathematical methods include correlation analysis, analysis of variance, and normal distribution. The principle of the correlation analysis method is to extract 100 sample pixels to generate the NDVI time profile curve and calculate the average [26]. Then, the correlation coefficient of 100 pixels is calculated to set the threshold for rice extraction. Then, the symbol test method is used to evaluate the difference between each pair of data from two related samples to compare the significance of the two samples. The variance analysis method uses multitemporal image data to calculate the time series curves of the vegetation index and calculate the standard deviation and variance of the vegetation index in each pixel, and then determines the threshold range by Formula (2). If the pixel value falls within the threshold range, it is determined as a paddy rice pixel [27]. The normal distribution method has the following assumption: the probability distribution function (PDF) of the land cover type follows a normal distribution [28]. We use the mean and standard deviation of each land cover type to define its normal distribution function, and two parameters are obtained from the training data set. The key to correctly distinguishing one specific land cover type is to minimize the overlaps between the target and the neighboring ordinary PDFs. For two land cover types L1 and L2, assuming L1~N($\mu_1$, $\sigma_1^2$) and L2~N($\mu_2$, $\sigma_2^2$), then the intersection between L1 and L2 is calculated by Formula (3).

$$V_{mean} - (nS) < x_1 < V_{mean} + (nS), \tag{2}$$

$$x_2 = \frac{\sigma_1 \mu_2 + \sigma_2 \mu_1}{\sigma_1 + \sigma_2} \tag{3}$$

where $V_{mean}$, $n$, $S$, $\mu$, $\sigma$, $x_1$, and $x_2$ are the average of the variance of the paddy rice field, the maximum distance from the standard deviation, the standard deviation of the variance

of the paddy rice field, and the average variance of the image to be classified, the mean of each land cover type, the standard deviation of each land cover type, the intersection between two land cover types. Generally, the two land cover types can be thought separable if $x_2$ is outside of $[\mu - \sigma, \mu + \sigma]$.

Chen used statistical methods to classify double-cropping paddy rice in Taiwan [26]. In addition, this paper also compared the accuracy of different smoothing methods with different NDVI time curves. Studies have shown that classification methods based on empirical mode decomposition (EMD) filtered data produce better classification results than wavelet transform. Nuarsa et al. [27] used the method of variance analysis and MODIS images to extract paddy rice from Bali, Indonesia. The results were good, and the kappa coefficient reached 0.8371. Wang et al. [28] used a normal distribution to process the threshold value of the vegetation index curve for paddy rice extraction in the eastern plains of China. This method was mainly applied to single-season rice. This method is only applied to the key phenological phase images of paddy rice growth. In addition, some studies have used the difference in NDVI during the critical phenology period to define the threshold for paddy rice mapping [12]. Liu et al. [29] proposed a subpixel method that used the relationship between the coefficient of variation (CV) of the LSWI and the planting fraction to estimate the planting fraction of paddy rice. The new method calculated the scale of paddy rice area based on the CV of the LSWI determined for pure water bodies and upland pixels, which can be automatically obtained from the MCD12Q1 land cover product. The overall accuracy was 88%.

The overall accuracy of this method is greater than 85%, and the kappa coefficient is greater than 0.7. The method has the advantages of simple principles and easy operation. The disadvantage is that the applicability of cloudy areas needs to be investigated. Mixed pixels and boundary effects will reduce the classification accuracy. Furthermore, it remains to be studied whether the accuracy of the method will be improved under the conditions of improved image spatial resolution, extended time series, and large-scale research areas.

The use of the phenology algorithm began in approximately 2000. Xiao et al. discovered the characteristics of the vegetation index and conducted paddy rice extraction studies in large areas such as South Asia and central and southern regions [30,31]. The results were good and showed the effectiveness of the phenology algorithm in paddy rice mapping. The previous method has some drawbacks. For example, the critical time window for paddy rice growth is obtained based on a large amount of agricultural phenology data. Incomplete agricultural phenology data in some areas will hinder the implementation of this method.

In recent years, paddy rice mapping methods have been continuously improved. The improvement is reflected in the use of high-resolution data sources, the increase in the complexity of the study area, the study of long-term sequences, and the increase in auxiliary materials (phenology information, other land cover masks, etc.). First, we will discuss high-resolution data sources. Previously, the MOD09A1 MODIS product was mostly used, but it has a spatial resolution of 500 m. For precision agriculture, there will still be mistakes. Subsequent studies used Landsat images and HJ-1A/B with a spatial resolution of 30 m, and Sentinel-2 with a spatial resolution of 10 m [10,11,14,28,33,40]. The accuracy has been further improved. Other studies have considered the issue of temporal resolution. MODIS and Landsat data have been integrated, and these data were then combined with a phenology algorithm for paddy rice mapping [10,32,41]. Second, the complexity of the study area also has an impact. Early studies were mostly concentrated in South Asia and other regions, and summer rainfall was mostly taken into consideration. With the expansion of paddy rice in Northeast Asia, the research area moved northward [14,33,34]. Compared with South Asia, the impact of early spring snowmelt should be considered due to the climate of the northeast region. Some scholars have studied the changes in the area of paddy rice in high temperature disaster areas [35]. Initially, research focused on paddy rice extraction in a specific area in a certain year to verify the accuracy of the

algorithm. Subsequent related studies focused on long-term sequence studies to study the expansion of paddy rice fields and changes in the planting area [14,33,36]. Finally, the increase in auxiliary information should also be considered. Some recent studies have attempted to use surface temperature or air temperature to define the time window that defines the temperature that should be reached during the key growing period of paddy rice, effectively excluding the effects of summer rainfall and early spring snow melt on monitoring [10,11,14,33–38]. Other relevant mask data also include cloud cover, snow cover, seasonal water cover, evergreen vegetation, and DEM. The algorithm flow chart is shown in Figure 2. The statistical data brought by the state's advocacy for refined agriculture have greatly facilitated the extraction of paddy rice. In addition, some studies have used the phenology algorithm to extract the spatial distribution of paddy rice with different planting intensities, which showed the potential of the phenology algorithm in describing two- and three-season paddy rice [39]. Some studies have added the results of field spectrometer measurements on the basis of previous optical remote sensing data to verify the changes in the rice vegetation index curve [32].

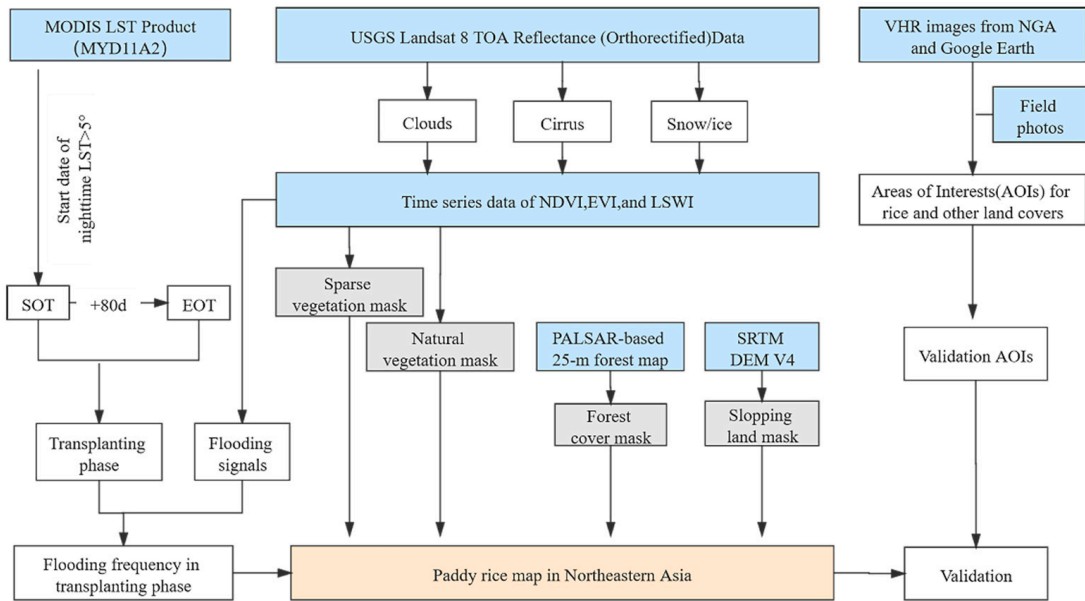

**Figure 2.** The workflow for phenology and pixel-based paddy rice mapping, major modules include time window determination of the rice transplanting phase (starting point: SOT, ending point: EOT), Landsat data preprocessing, phenology- and pixel-based mapping for non-cropland masks and paddy rice flooding, validation based on the areas of interest (AOIs) from very high-resolution (VHR) images and field photos [33].

The accuracy of rice mapping methods based on phenology is usually high, exceeding 80%. The advantage of this method is that it is suitable for long-term sequence dynamic analysis and large-scale observations. Based on phenological observations, the rice growth period can be accurately identified, reducing the need for data processing work. The principle of the method is simple and operable. The disadvantages of this method include errors in the cloud coverage area, mixed pixel problems, and limited observations over scattered landscapes. The recognition accuracy of clouds is high, but the recognition accuracy of cloud shadows is usually low. Because cloud shadow pixels usually meet the threshold of LSWI–EVI > 0, they may affect paddy rice field mapping. In addition, the inundation of the surface caused by extreme precipitation events can also affect paddy rice mapping [10].

### 4.1.4. Object-Based Image Analysis

There are three key steps of the object-based image analysis method: (1) segmentation of generated image object; (2) determination of features based on feature extraction of

objects; and (3) classification (multiple classification methods). Su [42] focused on using phenology to classify paddy rice under the object-based image analysis framework. The main purpose of this framework is to study the applicability of phenology in the localization of paddy rice based on object-based image analysis. The image segmentation is performed using the multiresolution segmentation algorithm in eCognition software. Then it is classified based on the neural network classification method. Singha et al. [43], in order to improve the segmentation quality, improved the fusion criterion on the basis of the commonly used fractal network evolution method, and a new segmentation algorithm was proposed. An unsupervised scale selection method was proposed to determine the optimal scale parameters for image segmentation, and to automate the process of determining scale parameters. After segmentation, geometric, spectral and texture features were extracted and input into the subsequent classification process. Paddy fields and non-paddy fields were classified by a random forest classifier. Zhang et al. [44] also performed image segmentation by using the multiresolution segmentation algorithm in eCognition 9.0 software. The prototype objects were classified by using the random tree (RT) classifier.

The accuracy of this method is generally better than that of other methods. The overall accuracy is over 90%, and the kappa >0.82. The advantages of this method are that geometric information, texture information and spectral information can be used simultaneously to improve the extraction accuracy, and the method analyzes objects by integrating neighborhood information rather than pixels, which will reduce the "salt and pepper" effect when rendering heterogeneous landscapes to classify paddy rice fields more accurately. Object-based image analysis shows advantages in identifying broken paddy rice fields. The disadvantage of this method is that the accuracy of the method is related to the accuracy of data, cloud pollution, spatial resolution, and processing of mixed pixels. In addition, image segmentation is still a challenging problem. Improving the quality of image segmentation is a key factor.

### 4.2. Microwave Remote Sensing-Based Mapping Methods

The use of a microwave source is a second type of mapping method for paddy rice. The first spaceborne synthetic aperture radar (SAR) sensor for paddy rice mapping used data from the European Remote Sensing Satellite 1 (ERS-1), which showed good results [45]. These groundbreaking studies were often limited to small-scale studies due to a lack of high-quality ground truth images, single polarization, or large data volumes. Subsequent research began to focus on using multiple SAR sensors to improve rice mapping over large land areas, and ERS-1, ERS-2, and RADARSAT were used to test various algorithms. Recent research included RADARSAT-2 data, combined optical and SAR data, object-oriented crop mapping, and Sentinel-1 C-band SAR data. Sentinel-1 satellite data can be obtained freely and openly all over the world, further promoting large-scale rice monitoring operations using radar data.

The main advantage of microwave remote sensing is theoretically the ability to acquire images under any weather conditions, such as cloud cover, rain, snow, and solar irradiance. In most cases, paddy rice cultivation is carried out during the rainy season when overcast and rainy weather prevails. Therefore, the radar image collected by the microwave sensor is an excellent image source for mapping paddy rice areas. In the growth process of paddy rice, the time series change in the radar backscatter coefficient is the key factor to distinguish paddy rice areas. The characteristic of the backscattering coefficient in the growth stage of paddy rice is that in the nutrition and reproduction stage, the backscattering increases continuously until it reaches the maximum at the heading stage. With the development of paddy rice phenology, stems elongate and leaf area, plant water and biomass increase. These changes increase the area available for radar wave reflection, leading to an increase in measured backscatter. After the heading stage, due to plant water, leaf area and biomass begin to decrease, the aforementioned scattering effect is reduced, resulting in a decrease in SAR backscatter. This time backscattering behavior is illustrated in Figure 3, which is based on multiyear advanced synthetic aperture radar (ASAR) wide swath mode (WSM) time

series data and shows the SAR backscattering behavior with triple-cropped rice growing stages.

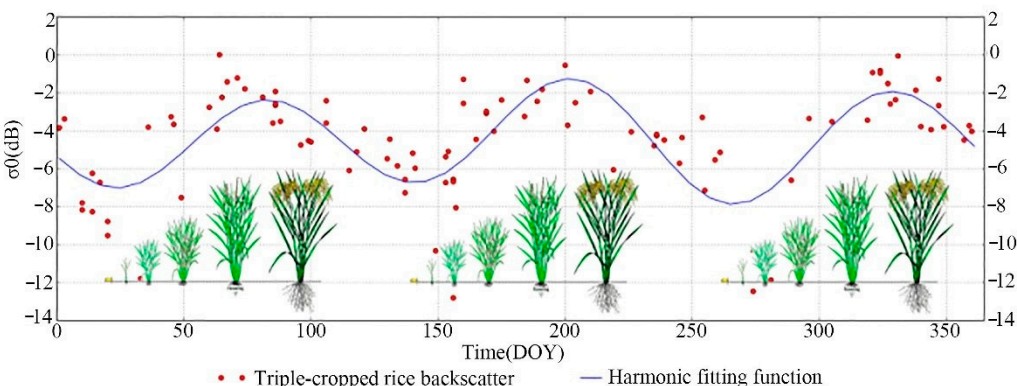

**Figure 3.** SAR backscatter behavior with triple-cropped rice growing stages based on a multiyear ASAR WSM time series [49].

### 4.2.1. Empirical Model

The earliest method of applying radar data to paddy rice mapping was to observe the changes in the backscattering coefficient during the paddy rice growth cycle to establish an empirical model. The principle of this method is to establish a mathematical formula based on the change in the backscattering coefficient during the paddy rice growth cycle, determine the threshold, coefficient and other parameters, and extract and map the paddy rice according to the parameters. In 2001, Shao et al. [46] investigated the backscattering behavior of paddy rice throughout the growth cycle, and paddy rice monitoring and extraction were carried out according to its characteristics. An empirical model of the paddy rice growth cycle and backscattering coefficient was established with an accuracy of 91%. However, the disadvantage of this method is that it has a single channel and a fixed angle of incidence. It is difficult to estimate multiple parameters for a target, and the target recognition ability needs to be strengthened.

In the past few years, with the advancement of algorithms and the diversity of data, empirical models have also been developed. In 2011, Bouvet used multitrack wide-swath data sets combined with former methods, using temporal backscatter changes as a classification feature for mapping. Compared with the previously used single-track narrow-swath data sets, this method can significantly increase the observation frequency and the size of the mapping areas [47]. The disadvantage is that the establishment of the empirical model must use the existing detailed land cover data to establish the equation and determine the classification threshold. When no ground information is available, the values in the previous literature are used, and there will be errors.

Radar data contain band information of different frequencies, and most previous methods have used C-band information. In 2018, Hoa et al. [48] used COSMO-SkyMed X-band SAR data to analyze the changes in the SAR intensity over time for short- and long-period paddy rice varieties and field seeding periods in the Anjiang region of the Mekong Delta. First, based on the survey data, a comprehensive analysis of the characteristics and cultivation techniques of paddy rice crops in the region was carried out. Then they analyzed the differences of backscattering intensity between paddy rice and other land cover types in this area under vertical transmission/vertical receive (VV), horizontal transmission/horizontal receive (HH) and HH/VV, and obtained indicators closely related to paddy rice mapping. Paddy rice fields were distinguished from other LULCs, and indicators derived from HH polarization could be used to map other LULCs (water, forests, and built-up areas). These maps can be used as auxiliary data to improve the accuracy of the results. The results showed that, due to the vertical structure of the paddy rice plants, this ratio was a good indicator for paddy rice field mapping. Vertically polarized waves are more attenuated than horizontally polarized waves, so the ratio of the

backscattered intensity of HH and VV is higher over paddy rice than most other land cover types. The accuracy of the paddy rice planting area has been found to be as high as 92%. However, for provincial and national surveying and mapping, the coverage of satellite data sources is a limitation. In this case, this method is more suitable for large coverage data with frequent repetition cycles.

The advantage of this method is that the idea is relatively simple, and the threshold can be developed for analysis and extraction after determining the threshold according to the data extraction features of the long-term sequence. The disadvantage is that this method depends on long-term observation data and is limited by data availability. The temporal resolution must meet the needs of the paddy rice growth cycle. On the other hand, there must be accurate prior knowledge in the study area to facilitate the establishment of equations and verification of results. The universality of the method is also limited. The backscattering coefficients of paddy rice will show different characteristics in different regions, and the parameters will change.

### 4.2.2. Machine Learning

In recent years, the machine learning method has mostly been used for paddy rice mapping based on radar data, which extracts the eigenvalues of the backscatter coefficient and inputs these values into the classifier for paddy rice mapping. Classification models mainly include traditional machine learning models (DT, SVM, RF) and deep learning models such as CNN and recurrent neural network (RNN). There are similar methods based on optical remote sensing data. The principles of these two methods are similar. The difference is the input of the training sample. The former method's input data include optical images and vegetation index curves. The latter inputs the backscatter coefficient value extracted after radar image processing. Both methods will consider the input of phenology information and texture feature information to improve the accuracy of the results.

In 2015, Nguyen et al. [49] normalized the data collected over many years and multitrack SAR with a statistical method and then classified it through a knowledge-based DT method. This study obtained an overall accuracy of 85.3%, kappa coefficient of 0.74. He et al. [50] used the backscattering coefficient and its combination with phenological information as inputs to the DT classifier for classification, and HH/VV, VV/VH, and HH/VH ratios were found to have the greatest potential for phenology monitoring. The overall accuracy level of 86.2% was obtained in this study. In March 2017, the Sentinel-2 satellite was launched. The following radar data research mostly used Sentinel-2 data as the data source. In 2019, the temporal behavior of the SAR backscattering coefficient over 832 plots containing different crop types was analyzed. Using the derived metrics, paddy rice plots were mapped through two different methods of DT and RF. The overall accuracy is high; the former has an overall accuracy of 96.3%, and the latter is 96.6% [51]. In addition, researchers have done further research on the combination of SAR and deep learning. Wang et al. [55] used crowdsourced data, Sentinel-2 and DigitalGlobe images, and CNN to map crop types with an overall accuracy of 74%. Secondly, we consider the RNN model. The commonly used method in the RNN model is the long short-term memory (LSTM) model and its improvements, such as bidirectional LSTM (Bi-LSTM). Researchers use the model and backscatter coefficient time series data to achieve a paddy rice map. Compared with traditional machine learning models [56–58], the research results show that the overall accuracy of RNN model results is 95%, and the accuracy of deep learning models is better than traditional machine learning models. One point to mention is that different radar polarization methods have different results. In 2016, Hoang and others used SAR to map paddy rice crops in the Mekong Delta [52]. This study used two methods of single polarization, dual polarization and full polarization to map paddy rice, and the classification accuracy increased with the complexity of the polarization method. In 2018, Lasko et al. used a random forest algorithm and Sentinel-1 radar time series images to draw a double-season and single-season paddy rice map of Hanoi, Vietnam, with resolutions of 10 and 20 m, respectively, using VV and VH polarization methods [53].

The overall accuracy of the 10-meter VV and VH polarization was the highest (93.5%). Subsequent research can focus on the comparison of multipolarized SAR data with different frequencies (C, X, L) to obtain the optimal combination.

Moreover, in 2017, Clauss et al. [54] proposed a method of drawing paddy rice planted area maps using Sentinel-1 time series using superpixel segmentation and phenology-based DTs. Superpixel segmentation is the establishment of a spatially averaged backscattering time series, which has the characteristics of robustness to speckles and reduces the amount of data to be processed. However, the classifier depends on the phenology-based empirical thresholds of the research site. If this method is applied to other regions, it is recommended to adapt the threshold parameters.

The advantage of this method is that paddy rice mapping is carried out by means of machine learning, feature extraction is performed using a large amount of data, and the overall accuracy is improved. However, this method relies on the input of training data to determine the parameters, and different regions will result in different parameters. The completeness and diversity of the training data determine the accuracy of this method.

In general, the accuracy of extraction algorithms based on optical remote sensing improves with the improvement of the method and the improvement of data quality. Most of the time series studies focus on annual series changes. The study areas are relatively large, covering the national scale, and these data generally have high spatial resolution. From the original spatial resolution of 500 m to the current spatial resolution of 30 m, it has been continuously improved, and the characteristics of the data are mainly large-scale. The research mainly focuses on the dynamic changes in the paddy rice area and the changes in the centroid of the paddy rice planting in the region. Extraction algorithms based on microwave remote sensing and rice monitoring based on the backscattering coefficient generally have high accuracy, approximately 90%, and the time series are mainly concentrated on the monthly scale. The study area is mostly within the province and city, with a resolution of 10 m, and its largest advantage is the tropics, where cloudy and rainy conditions dominate.

### 4.3. Integration of Optical and Microwave Remote Sensing-Based Mapping Methods

Optical remote sensing images and microwave data have their respective advantages. To improve the data accuracy, integrated analysis using both methods is essential. The integration methods are mainly the following, and the accuracy of the results is higher than that of a single data source.

#### 4.3.1. Complementary Method

The main principle of this method is to first obtain the rice extraction layer with optical remote sensing or radar data and then supplement the layered data from another data research institute or use these two data sources as the input layer for the classifier for a comprehensive analysis. This method mainly includes the following complementary methods: (I) The phenological information is determined based on the optical data. Radar images are collected based on phenological information for rice mapping. (II) The optical features of rice and the radar features are input into the classifier together for rice mapping. (III) The results are output separately based on the two data sources. The intersection of the two results is treated as the final result.

Using the first type of method, Asilo et al. [59] extracted paddy rice planting information based on MODIS and SAR images, and the results indicated that MODIS can be used to guide SAR image acquisition and planning to a large extent. Torbick et al. [60] conducted a large-scale paddy rice extraction experiment in Myanmar. In this study, Landsat 8 and other data were used to generate a large-scale land cover map, and then the radar image backscatter coefficient was used to create a detailed range of paddy rice masks. Using the second type of method, Mansaray et al. [61] focused on rice extraction in Shanghai, China. By combining the backscatter coefficient of the radar image with the vegetation index, the decision-making classification method was used to extract rice. Tian et al. [62] used the

characteristics of the backscattering coefficient and NDVI to enhance image information and combined this information with k-means unsupervised classification to determine the rice area of Poyang Lake in China. Fiorillo et al. [64] used Sentinel1 and Sentinel-2 data to extract rice spectra and backscatter coefficient features in degraded areas, and input them to the RF classifier together. The combination of Sentinel-1 and Sentinel-2 dense time series provided reliable predictors for RF classification, and the results were good. The overall accuracy is greater than 80%. Chen et al. [65] applied this method in a multi-cloud area and used the Google Earth Engine (GEE) platform. Overall accuracy is 66%. In the third type of method, Guo et al. [63] proposed an optical SAR collaborative paddy rice extraction method. The characteristics of rice growth were collected and analyzed under optical images and SAR for classification. Based on the rule that pixels with one of the classification results as rice are classified as rice, a collaborative fusion method was developed. In one area of Australia, the overall accuracy rate reached 94.7%. Ramadhani et al. [16] first extracted rice using Sentinel-1 and -2 and MODIS data, respectively combined with the SVM classification method, and then fused the two results to generate a multitemporal rice map. The advantage of this method is that it combines the advantages of two data sources. This method also effectively avoids the defects of a single data source. To a certain extent, the accuracy of the results has been improved. However, shortcomings still exist. For example, the spectral similarity of different crops is one shortcoming. Both data sources suffer from this problem. Whether data fusion effectively avoids this problem remains to be studied.

### 4.3.2. Comparison Class Method

The principle of this method is mainly based on different data combination methods, different classification methods, and the results of different regions to obtain the optimal combination of methods for paddy rice extraction. For example, the results of the same data input to different classifiers can be compared, and the results of radar data in different polarization modes combined with the same optical index can also be compared. Comparisons between pixel-based classification and area-based classification have also been conducted.

Onojeghuo et al. [66] took the Sanjiang Plain in northeast China as the research area, utilized NDVI images and dual-polarization (VH/VV) SAR as input data, and used RF and SVM machine learning classification algorithms to perform paddy rice mapping. The results showed that the RF algorithm applied to multitemporal VH polarization and NDVI data produced the highest classification accuracy (96.7%). Zhang et al. [67] first performed image preprocessing on Google Earth Engine (GEE) and combined the pixel-based classification results with object-based segmentation results to output a paddy rice area map. The combination of the two methods eliminated the noise that is common in medium- and high-resolution pixel classification and brought the rice planting area closer to official statistics. As a result, rice maps with a resolution of 10 m were established in Heilongjiang, Hunan and Guangxi provinces of China, with a total accuracy of approximately 90%. In the same year, Yang et al. [68] combined the characteristics of multiple watershed and mountainous areas in Wuhua County, South China, and used region-based and pixel-based methods to map the paddy rice planting area. The results showed that the accuracy of the area-based method was 1.18% higher than that of the pixel-based method (91.38%). The area-based method mainly eliminates the influence of speckle noise. Park et al. [69] classified paddy rice based on different data input combinations (original image, vegetation index, backscatter coefficient) combined with RF and SVM. The results showed that the fusion optics and SAR data had the highest accuracy. In this study, the Paddy Rice Mapping Index (PMI) was established based on the spectral and phenological characteristics of paddy rice, which could be used to extract paddy rice over a large area.

In fact, this kind of method is complementary to the first method. Here, we focus on the comparison between different methods. Researchers can choose the appropriate method according to their own research needs. For the advantages and disadvantages,

please refer to the advantages and disadvantages of the first method. There is limited literature on data fusion, and such studies have only appeared in recent years. These studies catered to the development trend of multisource data. Therefore, the problem of how to achieve the best fusion effect will be a focus of future work.

## 5. Discussion

### 5.1. Method Evolution Trend

From 2010 to 2020, paddy rice mapping methods were continuously innovated, following the development of science and technology (Figure 4). Previously, paddy rice mapping methods mainly consisted of images combined with simple classification methods, vegetation indices, etc. In 2010, the main method was still the phenology algorithm. Studies performed algorithm verification with different data sources in different regions. In 2011, radar data began to enter the field of paddy rice mapping. The emergence of radar data brought opportunities for paddy rice mapping in cloudy and rainy areas. The further development of the method improved the mapping accuracy of paddy rice by combining the method with computer technology. These methods include object-oriented, cloud computing, deep learning, and machine learning. GEE is a platform for online visualization computing analysis and processing to use Google's abundant computing resources for large-scale geospatial data processing [15]. The collocation of remote sensing methods, data, and processing infrastructure will help create high-resolution remote sensing products that cover a large scale. With the continuous launch of satellites, the functionality of satellites has been continuously enhanced, and the resolution of satellite data has also been continuously improved. Researchers have begun to focus on multisource data fusion, large-scale paddy rice extraction, and surface temperature data to improve phenology algorithms. The future development direction is actually very clear. The first is the use of Cubsat, GF series satellites, and satellite fusion data sets (harmonized Landsat and Sentinel-2(HLS) data set [70]). High-resolution satellite data provide a reliable foundation for mapping methods. The use of data sets can reduce the calculation procedures of researchers and further promote the development of mapping methods. Secondly, the combination of computer technology for image processing greatly improves the data processing efficiency and accuracy to a certain extent. In addition, the development of the unmanned aerial vehicle (UAV) has also provided a direction for paddy rice mapping and is suitable for the development of fine agriculture.

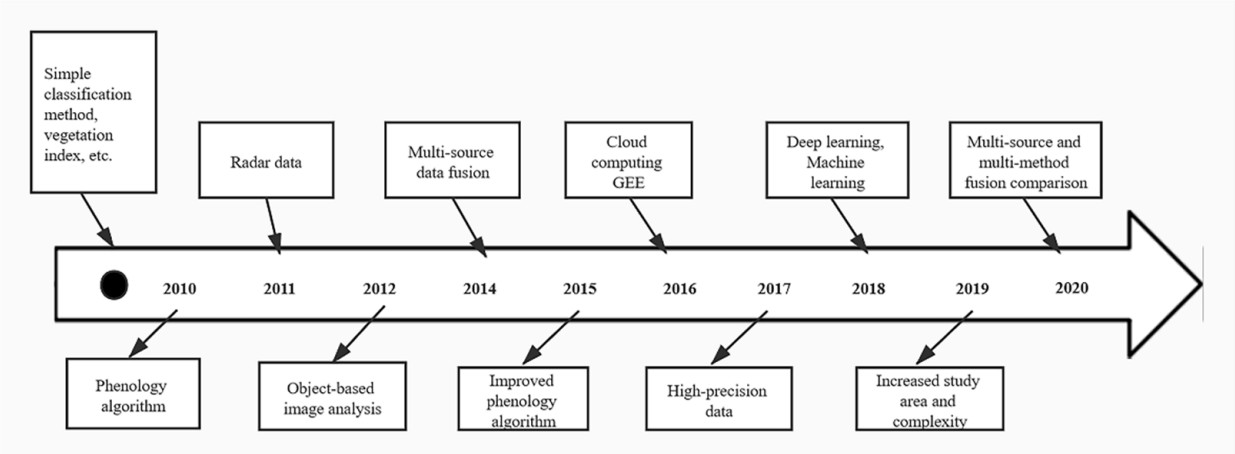

**Figure 4.** Evolution of methods.

### 5.2. Research Challenges

Although the methods have been continuously improved in recent years and the accuracy of paddy rice mapping has been continuously improved, challenges remain.

The discussion of previous methods mentioned that cloud cover is a major problem in paddy rice mapping. However, with the continuous improvement of methods and data sources, the impact of cloud cover has been reduced. The fusion of multisource data, which involves the combination of MODIS and Landsat data, has filled the gaps in the time series of the area covered by clouds [10,32]. The emergence of free SAR data and its cloud-penetrating characteristics also effectively solved the problem of cloud cover. Some studies have established time series models to eliminate the effects of cloud cover [36].

The problem of data verification remains. As previously mentioned, the improvement of statistical data in various regions and the emergence of high-resolution images have improved the quality of data verification. However, verifying the data remains a challenge because, based on artificial statistics, you cannot be sure of their accuracy. However, this challenge may gradually be overcome. With the continuous development of paddy rice mapping methods, a large number of research results have emerged, and there are overlapping areas in the study area; you can refer to the literature for data and method comparisons.

Therefore, the greatest problem is the versatility of the method. In different research areas, the threshold settings used by the methods are different, and different research areas may have different results; thus, a universal method is lacking. Most of the methods mentioned in this paper rely on the vegetation index and the time-varying curve of the backscattering coefficient for research. Machine learning and deep learning rely on training samples to improve the classification accuracy. In different regions and under different climatic conditions, the growth cycle of paddy rice will change, and the characteristic curve will change, which will change the number of features. Training samples also need to be re-extracted. There are also studies on using pixel segmentation algorithms based on different regions to jointly take eigenvalues to verify the generalizability of the method. However, the representative area in the study remains still to be studied [54].

## 6. Conclusions

Through a review and analysis of paddy rice remote sensing mapping methods applied over the past 10 years, the paddy rice mapping methods are divided into three categories according to the data source and then subdivided according to the principle of the method. Overall, there are many studies on paddy rice mapping using optical images, with MODIS, Landsat, and Sentinel-2 as the main data sources. With the emergence of Sentinel-1 data, research on the extraction of paddy rice based on radar data has gradually increased, effectively eliminating the problem of clouds and fog in optical images. The emergence of the concept of multisource data fusion has also brought good news to rice mapping, greatly improving the accuracy of rice extraction. Among these methods, there are classic methods and innovative methods. The best method in optical remote sensing is paddy rice mapping based on paddy rice phenology. Many studies have confirmed the applicability of this method in different regions, indicating the high overall accuracy. Innovative methods include the spectral matching method and threshold method using a combination of mathematical principles. Both of these methods rely on a time series graph of the vegetation index. The effects in cloudy and foggy areas need to be considered, and the effect may be improved by using a combination of the spatiotemporal data fusion models. With the development of computer technology in recent years, object-oriented and machine learning methods have emerged in the field of paddy rice mapping. In microwave remote sensing, the method combined with machine learning has high overall accuracy. When training the model, paddy rice phenology information and texture information can also be combined to improve model accuracy. The combination of optical remote sensing data and microwave data is a development direction for paddy rice extraction in the future. The advantages of the two types of data complement each other. Optical images can provide guidance for radar data, and radar data can provide assistance for optical data.

Combined with the above analysis, the following insight is obtained. One feature is multisource data fusion, which realizes rice mapping from a system perspective. The development of technology and the emergence of more accurate data sources with increased

spatial and temporal resolution provide new opportunities for rice monitoring. With the continuous innovation of algorithms and the continuous improvement of computing power, methods such as cloud platforms, GEE, and machine learning have emerged. Radar remote sensing images and optical remote sensing images can be effectively combined to better realize rice identification and monitoring. Integrated systems are the focus of future research. The second feature is the extraction of rice area under different planting systems. Most of the previous research focused on the accuracy of the algorithm. Most previous studies did not mention the rice area under different planting systems. Therefore, when paying attention to the extraction of rice and non-rice regions, it is necessary to focus on the analysis of the difference in yield caused by the difference in the paddy rice internal planting system. These results can provide powerful help for global food security. The third feature is to attach importance to issues such as global change and the ecological environment. Globalization is currently a major trend. On the basis of accurate paddy rice mapping, we must integrate global changes. Studies should pay attention to the environmental problems caused by rice growth, which will provide a better understanding of the response and adaptation of agricultural systems to global climate change.

**Author Contributions:** Conceptualization, Y.L., M.M. and R.Z.; methodology, Y.L. and R.Z.; formal analysis, R.Z.; writing—original draft preparation, R.Z.; writing—review and editing, Y.L. and M.M. All authors have read and agreed to the published version of the manuscript.

**Funding:** This research was funded by the National Natural Science Foundation of China, grant number: 41571419, 41830648, 41771453.

**Institutional Review Board Statement:** Not applicable.

**Informed Consent Statement:** Not applicable.

**Data Availability Statement:** No new data were created or analyzed in this study. Data sharing is not applicable to this article.

**Acknowledgments:** In this study, the papers were downloaded from the Web of Science and China National Knowledge Infrastructure, thanks to the retrieval and download services of the above websites.

**Conflicts of Interest:** The authors declare no conflict of interest.

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
