# Peer review of "Mapping Paddy Rice with Satellite Remote Sensing: A Review"

_sustainability, doi:10.3390/su13020503_

Round 1

Reviewer 1 Report

Zhao et al provided a literature review on using remote sensing data for paddy rice mapping. Considering the importance of paddy rice and the rapid development of applying remote sensing data for land cover classification, this study is interesting and necessary. However, the current form has several serious issues and lacks scientific innovations. Here are some comments that may be helpful to improve the manuscript.

  1. The title is about remote sensing, but the manuscript only talked about satellite remote sensing. Please narrow down the title or add relevant remote sensing literature review on other sources of remote sensing data.
  2. All figures are blurry. It is very difficult for readers to read figures. Please ensure that figures have enough resolution at least 300 dpi.
  3. Some new studies on using Cubsat or new satellite sensors should be included. 
  4. The review part on machine learning is also a little weak. Other methods such as RNN should also be reviewed.
  5. Captions for tables should also be improved and elaborated. 

Reviewer 2 Report

General comment:

This is an interesting paper. My main concerns are from their categorical approach. The authors’ divided the literature into three categories based on the input data used for their studies (i.e., optical, microwave, and integration of both satellite data), and then further divided each of the categories into several sub-categories based on implemented methods like the phenology method and machine learning. However, the concepts of terms they use are highly mixed. For example, supervised or unsupervised classification methods are just sub-categories of machine learning, and deep learning too. Further, for the phenology-based method, it’s worth to be noted as a key feature used in land cover mapping, but it is also can be considered just as one feature. In particular, phenology-related information can be used as one of the main inputs for any of the classification methods that is not comparable to machine learning. The authors should start with clear definitions of each term.

Line comments:

L97: Table 2; We know that revisit dates of a single Landsat image can be 16 days, but in the community, most of the people use a kind of integrated data, including multiple sources of Landsat archive. Therefore, the authors would present a kind of summary table of what is the temporal resolution of input data that was used in their study.

L377: It shows the seasonality of the spectral signal. Isn’t it phenology?

L545: From 2010 to 2020?

L565: Figure 4; It says “Cloud computing” has been introduced in 2016. What is the difference if this with “Cloud computing” in 2020?

Round 2

Reviewer 1 Report

The authors have revised the manuscript accordingly. I recommend this manuscript to be published on Sustainability.

Reviewer 2 Report

The authors have satisfactorily revised their manuscript. I have no further concerns that need to be addressed before publication other than a detailed read to catch typos.